# Successful Treatment with Patisiran in Amyloid Polyneuropathy Harboring His90Asn Mutation in the *TTR* Gene

**DOI:** 10.3390/brainsci14060519

**Published:** 2024-05-21

**Authors:** Vincenzo Di Stefano, Pietro Guaraldi, Francesca Giglia, Ilaria Cani, Antonia Pignolo, Luca Codeluppi, Paolo Alonge, Elena Canali, Giovanni De Lisi, Ada Maria Florena, Eugenia Borgione, Filippo Brighina

**Affiliations:** 1Department of Biomedicine, Neuroscience, and Advanced Diagnostic (BIND), University of Palermo, 90129 Palermo, Italy; niettapignolo@gmail.com (A.P.); alongep95@gmail.com (P.A.); filippobrighina@gmail.com (F.B.); 2IRCCS Istituto delle Scienze Neurologiche di Bologna, 40139 Bologna, Italy; p.guaraldi@isnb.it (P.G.); ilaria.cani@studio.unibo.it (I.C.); 3Unit of Neurology with Stroke Unit, S. Giovanni di Dio Hospital, 92100 Agrigento, Italy; francescagiglia@gmail.com; 4Neurology Unit, Neuromotor & Rehabilitation Department, Azienda USL-IRCCS di Reggio Emilia, 42122 Reggio Emilia, Italy; luca.codeluppi@ausl.re.it (L.C.); elena.canali@ausl.re.it (E.C.); 5Department of Health Promotion Sciences Maternal and Infantile Care, Internal Medicine and Medical Specialties, University of Palermo, 90133 Palermo, Italy; giovanni.delisi@unipa.it (G.D.L.); adamaria.florena@unipa.it (A.M.F.); 6Unit of Neuromuscular Diseases, Oasi Research Institute-IRCCS, Via Conte Ruggero 73, 94018 Troina, Italy; eborgione@oasi.en.it

**Keywords:** hereditary transthyretin amyloidosis, TTR, His90Asn, weight loss, patisiran, siRNA, gene silencing

## Abstract

Hereditary transthyretin amyloidosis (hATTR) is a multisystemic, rare, inherited, progressive and adult-onset disease, affecting the sensory-motor nerves, heart, autonomic function, and other organs. There are over 130 mutations known in the *TTR* gene. The His90Asn mutation has been previously reported in several reports, but its pathogenetic role is still debated. We report two sporadic cases of adult women with a heterozygous His90Asn mutation in *TTR* gene and neurological involvement extensively investigated. A typical Congo red-positive pathologic deposition of amyloid fibrils in the salivary glands was documented in one subject. Patients were successfully treated with patisiran with a good clinical outcome. These data support a pathogenetic role of His90Asn mutation in hATTR, and suggest early treatment in symptomatic carriers of His90Asn mutation.

## 1. Introduction

Hereditary transthyretin amyloidosis with polyneuropathy (hATTR-PN) is a multisystemic, rare, inherited, progressive and adult-onset disease, affecting the sensorimotor nerves, heart, autonomic function, and other organs [1]. hATTR is caused by mutations in the *TTR* gene, located in chromosome region 18q11.2–18q12.1. *TTR* gene mutations make the TTR tetramer unstable; as a consequence, a misfolded variant of TTR protein aggregates, generating amyloid fibrils, which accumulate as protein deposits in multiple organs and tissues [2]. hATTR is a devastating disease with a lethal outcome in a period of 2–15 years after the clinical onset in the absence of treatment. The approved drugs for the treatment of hATTR include TTR stabilizers and TTR mRNA silencers [3]. These recent new drugs for hATTR are very promising, and they might change the survival in patients with polyneuropathy [3]. Patisiran is a small, double-stranded interfering RNA encapsulated in a lipid nanoparticle, able to penetrate hepatocytes, where it selectively targets TTR mRNA, reducing TTR production, significantly improving the quality of life, nutritional status, and activities of daily living in hATTR patients [4,5]. Unfortunately, the diagnosis of hATTR is challenging, and it can be delayed for many years, thus postponing the start of effective treatments. Over 130 mutations in the *TTR* gene have been described, but some of them still lack a clear genotype–phenotype relationship. His90Asn mutation was first reported by Skare et al., who found that histidine at position 90 was replaced by asparagine, thus suggesting that intermolecular binding between hydrophobic polypeptide loops on the surface of transthyretin might lead to familial amyloidotic polyneuropathy [6,7]. Unfortunately, the genotype–phenotype correlations for this mutation have raised many questions in the following years. In a study from 1991, the researchers found that several asymptomatic Portuguese and German individuals carried His90Asn mutation without developing the disease [8]. Moreover, the only symptomatic patient of Portuguese ancestry identified presented a compound heterozygosis of *TTR* gene with His90Asn and Val30Met mutations; the authors excluded a pathogenetic role of His90Asn in that case, considering the Val30Met mutation the only pathogenetic contributor to the phenotype [8]. Unexpectedly, the same mutation was found in an American family of Italian origin with the hATTR phenotype [9]. The history continued three years later, when His90Asn mutations have been reported in hATTR patients in association with Glu42Gly, a known pathogenetic mutation found in Japanese patients with prominent neuropathy and cardiomyopathy [10]. Consequently, the authors concluded that His90Asn is a nonpathogenic variant of the *TTR* gene. Furthermore, a more recent paper reported the case of a young woman carrying His90Asn mutation with a cerebral hemorrhage in the absence of polyneuropathy [11]. In this study, we describe the case of two sporadic adult women with a heterozygous His90Asn mutation in *TTR* gene and neurological involvement extensively investigated and successfully treated with RNA-interference therapy.

## 2. Case 1

A 56-year-old woman from the South of Sicily came to our attention in March 2021 for nausea, vomiting, diarrhea with unexplained and progressive weight loss (up to 20 kg in 6 months), and distal paresthesia and numbness in the upper and lower limbs. She had history of smoking since she was twelve, and she was diagnosed with anxiety and major depression since the age of 35 with frequent hospitalizations. The family history was unrevealing for neuromuscular diseases, except for the patient’s mother, who had died at the age of 50 years from Guillain–Barré syndrome. Simultaneously with the onset of symptoms, the patient had lost autonomy requiring assistance for normal daily needs (washing, dressing, cooking, etc.) and further worsening of mood. After several visits to the emergency room, the patient was diagnosed with anorexia nervosa. Indeed, abdomen echography and CT of the chest and abdomen came back negative, and a gastroenterology consultation was also performed, but no disease was found to explain the weight loss and gastrointestinal symptoms. Laboratory tests, including electrophoresis, heavy chain immunofixation, oncological markers, antigangliosides, antineuronal antibodies, glycosylated hemoglobin, markers of hepatitis B and C, HIV, B1 and B12 vitamins, sexual and thyroid hormones, and cortisol were not-contributory. Also, a brain and lumbosacral MRI were unremarkable. Neurological examination revealed a severe sensory ataxia with a Romberg sign, mild distal weakness, distal hypoesthesia, hypopallesthesia and hyporeflexia from the knees down. Severe cachexia (weight: 35 kg, BMI: 14.6) was present with a blood pressure of 100/55 mmHg and a heart rate of 111/min. Nerve conduction studies demonstrated a sensory-axonal polyneuropathy in the upper and lower limbs (Table 1).

The contemporary presence of idiopathic sensory axonal neuropathy and gastrointestinal symptoms with unintentional weight loss raises the suspicion of hATTR. Hence, a genetic testing was immediately performed and demonstrated the c.328C>A (p.H110N) heterozygous variant in *TTR* gene, generally reported as His90Asn.

At first, the variant was considered of uncertain significance (VUS), because of the conflicting literature on this mutation. However, due to the high suspicion of hATTR, a biopsy from salivary glands was performed which showed eosinophilic and amorphous material in the perivascular and interstitial spaces. Congo Red stain confirmed the pathologic deposition of amyloid fibrils with the characteristic “apple-green” birefringence examined with the polarized microscopy (Figure 1). A cardiologic consultation with electrocardiogram and echocardiogram did not find any sign of cardiomyopathy except for tachycardia and bone scintigraphy with ^99mTc^hydroxydiphosphonate (Tc99-HMDP) did not show any cardiac uptake. N-terminal prohormone of brain natriuretic peptide (NT-proBNP) was 210 pg/mL, and T troponin was within normal limits. A cardiac MRI was performed after the diagnosis of FAP due to the suspect of a mixed phenotype, but it was unrevealing.

The patient promptly started patisiran with a brilliant response at 6 and 9 months of follow-up in absence of any side effects. The patient gained weight: +6 kg in the first 6 months (weight: 41 kg, BMI: 17) and +8.5 kg at 9 months follow-up (weight: 49.5 kg, BMI: 20.6); moreover, her symptoms of neuropathy significantly improved (from −7 points on NIS-W, from 19 to 12), she gained speed on 6MWT (from 148 to 285 mt, +93%) and quality of life (−35 points for Norfolk scores, from 95 to 60), as well as Karnofsky performance status (from 50 to 70%) (Table 2). Nerve conduction studies performed after 9 months of follow-up demonstrated an increase in the amplitude of sensory nerve action potential of the sural, median, and ulnar nerves (Table 1). At last evaluation in March 2023, she presents with unimpaired ambulation (FAP stage 1, PND score 1), stable body weight and improved distal sensory loss and muscle strength. A genetic study was performed in first-degree relatives, thus discovering the same mutation in two twins (33 years) and four sisters (46, 58, 60 and 62 years). All carriers of His90Asn mutation have been followed for a about two years, and they still have no sign or symptoms of hATTR.

## 3. Case 2

A 51-year-old woman with no family history for neuromuscular diseases came to our attention in 2018 for a feet burning sensation, leg cramps and subjective loss of balance in walking. Symptoms occurred few weeks after the removal of a skin lesion, which had revealed to be a malignant melanoma. Neurological examination was unremarkable except for a bilateral absent achilleus reflexes.

A first diagnostic work-up was performed, with an electrophysiological study disclosing a sensorimotor neuropathy with mixed axonal and demyelinating features. Due to the recent medical history, an extensive search for neoplastic causes including positron emission tomography was performed, and proved negative. The patient underwent a regular neurological follow-up, which documented worsening of symptoms two years after. Accordingly, the second electrophysiological study showed a slight progression of the neuropathic changes. Lumbar puncture was performed with normal results in the whole cerebrospinal fluid testing, but an extensive immunological blood assay disclosed positivity for anti-GT1a IgM, anti-GD1b IgM, anti GQ1b IgM antibodies. Additionally, in the meantime, an autoimmune thyroiditis emerged. Following the hypothesis of an autoimmune cause for the neuropathy, two trials with intravenous human immunoglobulin (IVIG) were then attempted at the full dose of 0.4 g/kg/die for five consecutive days for each trial; however, both of them provided no benefit and the treatment was therefore stopped. In the following year neuropathic symptoms further worsened and neurological examination revealed a slight distal leg weakness. Due to rapidly progressive sensorimotor neuropathy of undetermined cause with no benefit from IVIG, we performed a genetic testing for ATTR-PN, and the His90Asn mutation emerged. Cardiologic evaluation including electrocardiogram, echocardiogram and Tc99-HMDP ruled out signs of cardiomyopathy. The autonomic assessment with cardiovascular reflex test did not showed alteration in the autonomic control of cardiovascular system.

A treatment with patisiran was then started. After 12 months of treatment, the neurological examination showed improvement in the distal leg weakness (−7 point on NIS-W) and unimpaired ambulation (FAP stage 0, PND score 0) (Table 2). The patient also stopped losing wight (weight: 55 kg, BMI 19). She reported an improvement on her quality of life (−7 points for Norfolk scores; Karnofsky performance status 100%). Nerve conduction studies performed at that time showed a stability of the electrophysiological picture (Table 1). A cardiovascular reflex test confirmed no alteration of autonomic nervous system.

## 4. Discussion

The pathogenic role of the His90Asn mutation has been discussed by previous studies with conflicting results. We present two unrelated cases with heterozygous His90Asn mutation and hATTR phenotype in the absence of any other pathogenetic mutation in the *TTR* gene. All the exons of the *TTR* gene have been carefully examined, but only the c.328C>A heterozygous variant in *TTR* gene was demonstrated. This variant has an allele frequency of 0.00040 from The Genome Aggregation Database (gnomAD), while it was absent in our 120 Italian individuals (normal and disease control subjects). Furthermore, it is classified as VUS by ACMG/AMP 2015 guidelines, meeting criteria PM1, and PP2_supporting and reported in The Human Gene Mutation Database—Professional 2022.1 (HGMD) as possible disease cause. In 3D modeling (Figure 2), the exchange from native amino acid to Asn90 not showed altered intramolecular interactions. However, since the asparagine side-chain can form hydrogen bond interactions with the peptide backbone, possibly this could lead to an abnormal protein conformation and, as a consequence, TTR protein aggregation.

The connection of this mutation with the phenotype is confirmed by the presence of amyloid deposits in the salivary glands in one case, together with a clear multisystemic involvement (peripheral neuropathy, weight loss, gastrointestinal symptoms). Also, a prominent and rapid response to the treatment with patisiran was documented in both patients. However, these cases raise several considerations. First, in our patients, the lack of cardiac involvement seems to indicate that His90Asn might be associated with a neuropathic rather than cardiologic phenotype. Of interest, this phenotype is quite different from the one described by Skare, in which cardiomyopathy was prominent in all carriers of the compound heterozygosis [10]. Hence, in the family by Skare, the presence of Glu42Gly mutation might explain the cardiologic phenotype, already described in Japanese FAP patients [10]; a possible explanation is that His90Asn, when coinherited with Glu42Gly, might carry to a more severe mixed phenotype, with milder phenotypes with higher variability and incomplete penetrance when encountered in isolation. 

Also, some considerations arise from one single case in which the authors questioned the possible role of the His90Asn variant, based on the description of the case of a man of 46 years with an “amyloidogenic phenotype”, represented by familiar cerebral bleeding, which did not follow a coherent segregation in the family [11]. However, there is very little evidence for cerebral involvement in hATTR, and we believe that this criterion should not be used to rule out the diagnosis in most cases [12]. We believe that the patient described, and his relatives might be considered in the pre-symptomatic phase of hATTR, like for the sons and sisters in our first case. Moreover, expert consensus has recently defined a “predicted age of onset of symptomatic disease” (PADO) to determine when carriers should start regular monitoring for hATTR [13]. PADO depends on the specific *TTR* gene mutation, the typical age of onset for that mutation, and the age of onset in other members of the proband’s family [13]. Hence, in our case, the estimated PADO is about 41–46 years. In this perspective, some cases described by previous studies seem to be too young to be considered clearly affected; this might explain some negative reports on His90Asn pathogenicity [11]. The same reasoning applies to the two sisters who carrying His90Asn mutation (the mother died, and the patient’s children did not undergo to genetic study for ethical problems).

Furthermore, some considerations arise from the clinical presentation and the prominent response to the therapy. The high discrepancy in phenotypic expression of the hATTR greatly contributes to a high rate of misdiagnosis, delay in diagnosis and, consequently, in the therapeutic approach, especially in patients without a known family history and with a late onset of clinical manifestations [14,15]. Surprisingly, anorexia was the most common misdiagnosis in a recent study on the genetic screening for hATTR, accounting for 30% of misdiagnoses [15]. Furthermore, a study employing machine-learning algorithms for the early diagnosis of hATTR with 93 patients from the Center and South of Italy has recently confirmed that unexplained weight loss and gastrointestinal symptoms are highly predictive for a diagnosis of hATTR [16]. Indeed, a clinical presentation with unexplained weight loss might be confused with behavioral eating disorders [15,17], especially in the case of mutations in the *TTR* gene causing autonomic dysfunction, diarrhea, and prominent gastrointestinal symptoms. This report highlights the importance to diagnose hATTR in patients with unexplained malnutrition early, to permit an early access to new disease modifying therapies, which are available for hATTR. Also, the two patients experienced a significant clinical improvement and restoration of a good quality of life after 9–12 months of treatment with patisiran. A recent study with bioelectrical impedance analysis supports weight gain and muscle reorganization in hATTR patients after 9 months of follow-up after the start of patisiran [18]. Hence, the pathogenetic potential of His90Asn variant should be reconsidered in the presence of symptoms of hATTR. Indeed, patients who carry of this mutation can benefit from effective therapies, thus halting the disease course and, in some cases, reversing clinical features [3,18]. However, the possibility that improved nutrition support, as well as significant weight gain, might have contributed to amelioration of neuropathy. Also, systematic assessment of dysautonomia was not performed, and might be overestimated by patients and clinicians. Nonetheless, even if very unlikely, the possibility of wild-type TTR amyloidosis cannot be ruled out in the presented cases as, in very rare cases, the deposition of both wild-type and variant TTR misfolded proteins had been reported. Finally, according to the ClinVar database, the sequence c.328C>A has eight classifications, with five considered benign, two conflicting interpretations of pathogenicity, and one uncertain significance. Minor allele frequencies across different ethnicities range from 0 to 0.008 (Ashkenazi Jewish), with an estimated global population frequency of around 0.000348. Further information on familial segregation and animal models is necessary to confirm the pathogenic variant. Further studies are needed to clarify genotype–phenotype correlations in His90Asn mutation; a careful evaluation of mutation carriers is necessary to determine the clinical characteristics, penetrance, and clinical phenotypes.

## 5. Conclusions

His90Asn mutation of the TTR gene may have a pathogenetic role in hATTR; hence, in carriers of His90Asn mutation a characterization of amyloid deposits may be useful to achieve a diagnosis of hATTR and obtain an early treatment.

## Figures and Tables

**Figure 1 brainsci-14-00519-f001:**
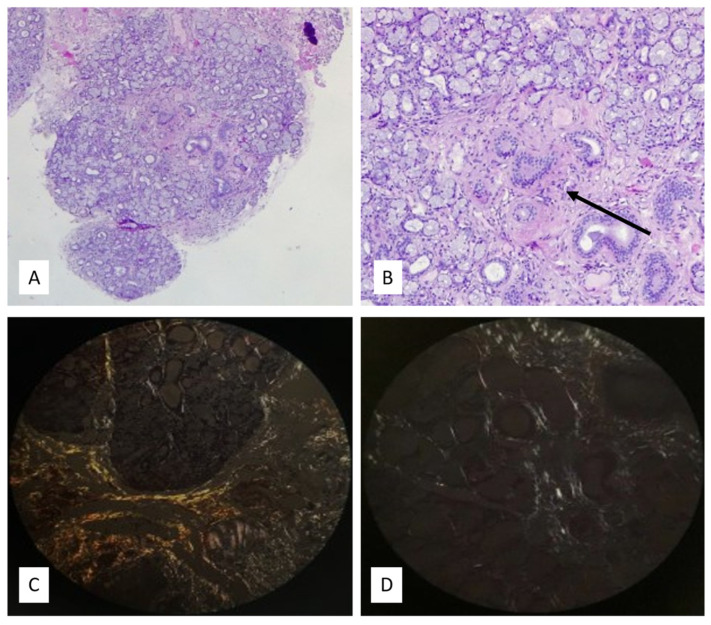
(**A**) Panoramic histologic view of the biopsy specimen: a fragment of minor salivary gland mainly composed of mucous secreting glands from patient 1. At low magnification, amorphous eosinophilic material is centrally observed (H&E stain, original magnification 40×). (**B**) The material mainly appears in the perivascular space and interspersed between the salivary ducts (arrow) (H&E stain, original magnification 100×). (**C**) Congo Red stain confirmed the amyloid composition of the amorphous material with the characteristic “apple-green” birefringence under the polarized light. (Congo Red stain, original magnification 200×). (**D**) Focal “apple-green” birefringence was also observed in the periglandular spaces. (Congo Red stain, original magnification 400×).

**Figure 2 brainsci-14-00519-f002:**
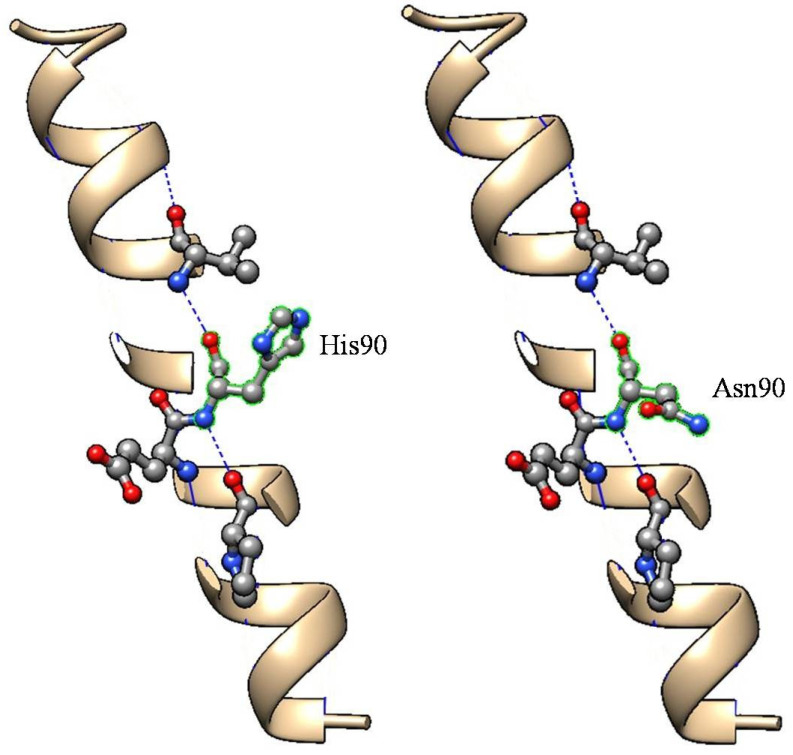
Comparison of 3D models for the normal and Asn90-TTR protein.

**Table 1 brainsci-14-00519-t001:** Summary of nerve conduction study at baseline and after therapy with patisiran. SAP, sensory action potential; VCS, sensory conduction velocity.

Sensory Nerve Study	Patient 1		Patient 2	
	Baseline	After Patisiran	Baseline	After Patisiran
Ulnar nerve right [digit V—wrist]				
SAP (uV)	3.0	17.0	4.7	9
VCS (m/s)	62	59	53	50
Ulnar nerve left [digit V—wrist]				
SAP (uV)	0.6	10.0	3.4	5.0
VCS (m/s)	54	66	55	48
Median nerve right [digit III—wrist]				
SAP (uV)	10.6	17.2	na	8.0
VCS (m/s)	55	63	na	38
Median nerve left [digit II—wrist]				
SAP (uV)	5.2	15.7	na	5.0
VCS (m/s)	55	61	na	48
Sural nerve right [ankle—foreleg]				
SAP (uV)	3.2	14.0	2.5	3
VCS (m/s)	50	49	30	44
Sural nerve left [ankle—foreleg]				
SAP (uV)	5.8	11.0	10.7	6.0
VCS (m/s)	59	60	41	50

**Table 2 brainsci-14-00519-t002:** Summary of clinical and physical evaluation at baseline and after therapy with patisiran.

Clinical Evaluations	Patient 1		Patient 2	
	Baseline	After Patisiran	Baseline	After Patisiran
Weight (kg)	41	49.5	54	55
BMI	17	20.6	18.7	19.0
PND score	2	1	1	0
NIS-W	19	12	18	11
6MWT	148	285	-	-
Norfolk scores	95	60	14	7
Karnofsky performance status (%)	50	70	90	100

Body-mass index (BMI); neuropathy impairment score (NIS-W); 6 min walking test (6MWT); polyneuropathy disability score (PND).

## Data Availability

The data underlying this article will be shared on reasonable request to the corresponding author. Data are not publicly available due to ethical reason.

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
