# Peer review of "Successful Treatment with Patisiran in Amyloid Polyneuropathy Harboring His90Asn Mutation in the TTR Gene"

_brainsci, 2024, doi:10.3390/brainsci14060519_

Round 1

Reviewer 1 Report (Previous Reviewer 1)

Comments and Suggestions for Authors

Since the pathogenic authentication for the missense sequence variant cannot be validated, I would suggest to modify the title into "Successful treatment with patisiran in the amyloid polyneuropathy patients harboring the sequence variant, p.His92Asn in TTR gene."

Author Response

Dear Editors and Reviewers,

Thanks for your comments. We would like to submit our revised version of the manuscript for possible publication in your journal. Here our point-to point answers to the reviewer questions:

Reviewer 1

Since the pathogenic authentication for the missense sequence variant cannot be validated, I would suggest to modify the title into "Successful treatment with patisiran in the amyloid polyneuropathy patients harboring the sequence variant, p.His92Asn in TTR gene."

A: We thank the reviewer for this suggestion. We modified the title according to the reviewer’s suggestion.

Hoping in positive feedback we look forward to hearing from you soon.

Kind regards,

Vincenzo Di Stefano

Reviewer 2 Report (Previous Reviewer 2)

Comments and Suggestions for Authors

The authors have improved the quality of their manuscript by adding discussion on some of the topics brought by the reviewers. It is quite complex to present new evidences towards the pathogenic role of some genetic variants and is markedly seen in the case of this manuscript. The c.328C>A (p.His110Asn) variant is currently classified by some important genomic labs as benign (including recent submission in the ClinVar database: https://www.ncbi.nlm.nih.gov/clinvar/RCV000014369/). The authors presented pathological evidence of amyloid deposition. It is not possible to completely exclude the presence of other amyloidogenetic origin of the deposits, such as with wild-type or primary amyloidosis, and it extremely rare to identify hATTR in patients without a positive family history of the disease (i.e., it is not possible to give certainty in the connection of the patient 1's mother with Guillain-Barre syndrome and any type of correlation with hATTR).  

One point that must be corrected by the authors is the citation of the variant along the manuscript - they use several ways ("p.His110Asn; p.His90Asn; His90Asn; his90asn") and must decide in only one description. 

Another point to be reviewed is that authors describe in lines 231 and 232 "Hence, similarly to other mutations in the TTR gene, his90asn should be considered pathogenetic in the presence of symptoms of hATTR" - this description is not correct, especially in the context of a large conflicting variant - I suggest authors remove this content. 

Other aspect is to discuss why they consider a variant that is not rare in several databases as potentially related to a rare neurological condition.

Author Response

Dear Editors and Reviewers,

Thanks for your comments. We would like to submit our revised version of the manuscript for possible publication in your journal. Here our point-to point answers to the reviewer questions:

Reviewer 2

The authors have improved the quality of their manuscript by adding discussion on some of the topics brought by the reviewers. It is quite complex to present new evidences towards the pathogenic role of some genetic variants and is markedly seen in the case of this manuscript. The c.328C>A (p.His110Asn) variant is currently classified by some important genomic labs as benign (including recent submission in the ClinVar database: https://www.ncbi.nlm.nih.gov/clinvar/RCV000014369/). The authors presented pathological evidence of amyloid deposition. It is not possible to completely exclude the presence of other amyloidogenetic origin of the deposits, such as with wild-type or primary amyloidosis, and it extremely rare to identify hATTR in patients without a positive family history of the disease (i.e., it is not possible to give certainty in the connection of the patient 1's mother with Guillain-Barre syndrome and any type of correlation with hATTR). 

One point that must be corrected by the authors is the citation of the variant along the manuscript - they use several ways ("p.His110Asn; p.His90Asn; His90Asn; his90asn") and must decide in only one description.

A: We thank the reviewer for this suggestion. We modified the citation of the TTR variant throughout the manuscript, referring as His90Asn.

Another point to be reviewed is that authors describe in lines 231 and 232 "Hence, similarly to other mutations in the TTR gene, his90asn should be considered pathogenetic in the presence of symptoms of hATTR" - this description is not correct, especially in the context of a large conflicting variant - I suggest authors remove this content.

A: We agree with the reviewer. We revised the sentence.

Other aspect is to discuss why they consider a variant that is not rare in several databases as potentially related to a rare neurological condition.

A: We thank the reviewer for this suggestion. Something like this happens for other mutations, like Val142Ile which is present in 3% of African-descendent population. Probably hATTR is not so rare as people may think. In Sicily we experienced relatively high incidence (Genetic screening for hereditary transthyretin amyloidosis with polyneuropathy in western Sicily: Two years of experience in a neurological clinic. Eur J Neurol. 2024 Jan;31(1):e16065. doi: 10.1111/ene.16065. Epub 2023 Sep 19. PMID: 37725003.).

Hoping in positive feedback we look forward to hearing from you soon.

Kind regards,

Vincenzo Di Stefano

Reviewer 3 Report (Previous Reviewer 3)

Comments and Suggestions for Authors I suggest the authors to make a structure predictions for the TTR variant. Otherwise manuscript seems fine.

Author Response

Dear Editors and Reviewers,

Thanks for your comments. We would like to submit our revised version of the manuscript for possible publication in your journal. Here our point-to point answers to the reviewer questions:

Reviewer 3

I suggest the authors to make a structure predictions for the TTR variant. Otherwise manuscript seems fine.

A: We thank the reviewer for this suggestion. We elaborated a predictive model of structure in figure. We also discussed possible modifications and its interaction with mutant TTR.

“In 3D modeling (Fig1), the exchange from native amino acid to Asn90 not showed altered intramolecular interactions. However, since the asparagine side-chain can form hydrogen bond interactions with the peptide backbone, possibly this could lead to an abnormal protein conformation and, as a consequence, TTR protein aggregation.”

Hoping in positive feedback we look forward to hearing from you soon.

Kind regards,

Vincenzo Di Stefano

Round 2

Reviewer 2 Report (Previous Reviewer 2)

Comments and Suggestions for Authors

The authors have properly discussed the additional points brought in the last review step. 

This manuscript is a resubmission of an earlier submission. The following is a list of the peer review reports and author responses from that submission.

Round 1

Reviewer 1 Report

Comments and Suggestions for Authors

The authors documented two cases of amyloid polyneuropathy, both featuring a heterozygous sequence variant, p.His90Asn. Following administration of patisiran, there was observable enhancement in electrophysiological parameters and overall quality of life.

Several significant concerns arise from this report:

1)      The primary issue revolves around verifying the pathological mutation, p.His90Asn. Given the absence of a positive family history in these patients, there's a lack of familial segregation between the sequence variant and the disease phenotype. While some genetic carriers in Case 1's family are older than the index case, they remain unaffected. Thus, it is dubious for authentication of this sequence variant as a pathogenic one.  

2)      Familial amyloid polyneuropathy typically manifests small nerve fiber involvement before affecting larger ones. Consequently, symptoms related to the gastroenteric and genitourinary systems usually precede sensory deficits in distal limbs. However, neither of the reported cases presented autonomic dysfunction prior to peripheral neuropathy onset, nor did they exhibit cardiac dysfunction.

3)      As axonal loss is a prominent feature in amyloid polyneuropathy, motor and sensory deficits typically follow a length-dependent pattern. Table 1 indicates that the sensory action potentials (SAPs) in sural nerves of both patients were larger than those found in ulnar nerves before treatment. Furthermore, the impairment of motor function has not been described.  

4)      Table 1 suggests an improvement in SAP for patient 1, possibly attributed to improved nutrition support and significant weight gain.

5)      The SAP in sural nerve of patient 2 did not show a significant increase after treatment.

6)      There is no data indicating an improvement in autonomic dysfunction for these two patients.

7)      According to the ClinVar database, the sequence c.328C>T (p.His90Asn) has eight classifications, with five considered benign, two conflicting interpretations of pathogenicity, and one uncertain significance. Minor allele frequencies across different ethnicities range from 0 to 0.008 (Ashkenazi Jewish), with an estimated global population frequency of around 0.000348. Further information on familial segregation and animal models is necessary to confirm the pathogenic variant.

Comments on the Quality of English Language

Improvements are needed in the English writing of the manuscript.

Author Response

Dear Editors and Reviewers,

Thanks for your comments. We would like to submit our revised version of the manuscript (Manuscript ID: brainsci-2969239 - Encourage Resubmission after Revisions) for possible publication in your journal. Here point-to point answers to the reviewer questions:

#Editor. The authors documented two cases of amyloid polyneuropathy, both featuring a heterozygous sequence variant, p.His90Asn. Following administration of patisiran, there was observable enhancement in electrophysiological parameters and overall quality of life. Several significant concerns arise from this report:

 #rev 1

1)      The primary issue revolves around verifying the pathological mutation, p.His90Asn. Given the absence of a positive family history in these patients, there's a lack of familial segregation between the sequence variant and the disease phenotype. While some genetic carriers in Case 1's family are older than the index case, they remain unaffected. Thus, it is dubious for authentication of this sequence variant as a pathogenic one.  

A: We partially agree with the reviewer. In fact, to complete a segregation analysis and to better explore these correlations with phenotype, it would be necessary to follow the family for decades, and we will do it, but in the meanwhile we can imagine that the evidence provided are already of interest for the literature. We do not think that the finding of positive amyloid deposition on biopsy nor sensory polyneuropathy and amelioration from treatment with patisiran may be random, especially considering that it is not just one case, but two different, unrelated cases with no genetic, or cultural connection. We strongly believe that this case significant add data to the literature even if further data are needed, of course.

2)      Familial amyloid polyneuropathy typically manifests small nerve fiber involvement before affecting larger ones. Consequently, symptoms related to the gastroenteric and genitourinary systems usually precede sensory deficits in distal limbs. However, neither of the reported cases presented autonomic dysfunction prior to peripheral neuropathy onset, nor did they exhibit cardiac dysfunction.

A: We do not agree with the reviewer. It can happen in some cases that small fibers are affected first, but it is not the rule. Any peripheral nerve can be affected by amyloid deposition. On the contrary, several studies have already demonstrated that urinary and gastrointestinal disturbances occur before motor impairment from severe large-fiber neuropathy (Kaku MC, Bhadola S, Berk JL, Sanchorawala V, Connors LH, Lau KHV. Neurological manifestations of hereditary transthyretin amyloidosis: a focus on diagnostic delays. Amyloid. 2022 Sep;29(3):184-189. doi: 10.1080/13506129.2022.2046557. Epub 2022 Mar 7. PMID: 35253562; Papagianni A, Ihne S, Zeller D, Morbach C, Üçeyler N, Sommer C. Clinical and apparative investigation of large and small nerve fiber impairment in mixed cohort of ATTR-amyloidosis: impact on patient management and new insights in wild-type. Amyloid. 2022 Mar;29(1):14-22. doi: 10.1080/13506129.2021.1976751. Epub 2021 Oct 11. PMID: 34632904).

3)      As axonal loss is a prominent feature in amyloid polyneuropathy, motor and sensory deficits typically follow a length-dependent pattern. Table 1 indicates that the sensory action potentials (SAPs) in sural nerves of both patients were larger than those found in ulnar nerves before treatment. Furthermore, the impairment of motor function has not been described.  

A: We do not agree with the reviewer. See also answer to question 2). Moreover, ulnar nerves usually display higher SNAP amplitude compared to sural nerves, hence the percentage of increment after patisiran may appear higher. In any case, these statistical considerations are meaningless given that we are talking about a single case report and not a large series of patients. In fact, the intrinsic variability of the recording method and the variability of response to treatment in the individual patient could explain this result. What matters, however, is not the magnitude of the response itself, but the presence of absence of a response, which constitutes indirect evidence of the pathogenicity of the His90Asn mutation.

4)      Table 1 suggests an improvement in SAP for patient 1, possibly attributed to improved nutrition support and significant weight gain.

A: We do agree with the reviewer. We added this consideration in the discussion.

“However, the possibility that improved nutrition support as well as significant weight gain might have contributed to amelioration of neuropathy.”

5)      The SAP in sural nerve of patient 2 did not show a significant increase after treatment.

A: In this case lack of improvement is not a proof against efficacy of patisiran, as from APOLLO study, a stabilization is expected and not amelioration. In some cases amelioration can happen (Adams D, Polydefkis M, González-Duarte A, Wixner J, Kristen AV, Schmidt HH, Berk JL, Losada López IA, Dispenzieri A, Quan D, Conceição IM, Slama MS, Gillmore JD, Kyriakides T, Ajroud-Driss S, Waddington-Cruz M, Mezei MM, Planté-Bordeneuve V, Attarian S, Mauricio E, Brannagan TH 3rd, Ueda M, Aldinc E, Wang JJ, White MT, Vest J, Berber E, Sweetser MT, Coelho T; patisiran Global OLE study group. Long-term safety and efficacy of patisiran for hereditary transthyretin-mediated amyloidosis with polyneuropathy: 12-month results of an open-label extension study. Lancet Neurol. 2021 Jan;20(1):49-59. doi: 10.1016/S1474-4422(20)30368-9. Epub 2020 Nov 16. Erratum in: Lancet Neurol. 2021 Feb;20(2):e2. PMID: 33212063).

6)      There is no data indicating an improvement in autonomic dysfunction for these two patients.

A: We agree with the reviewer. This was added in the discussion.

“Also, systematic assessment of dysautonomia was not performed and might be overestimated by patients and clinicians.”

7)      According to the ClinVar database, the sequence c.328C>T (p.His90Asn) has eight classifications, with five considered benign, two conflicting interpretations of pathogenicity, and one uncertain significance. Minor allele frequencies across different ethnicities range from 0 to 0.008 (Ashkenazi Jewish), with an estimated global population frequency of around 0.000348. Further information on familial segregation and animal models is necessary to confirm the pathogenic variant.

A: We agree with the reviewer. We added these considerations in the discussion.

Comments on the Quality of English Language

Improvements are needed in the English writing of the manuscript.

A: Manuscript was doublechecked for grammar and style.

Reviewer 2 Report

Comments and Suggestions for Authors

1.        I recommend authors to change the use of italics after the TTR gene symbol. The use of italics should be limited to the symbol (i.e., TTR, GSN, SPG11PMP22). This suggestion should be reviewed both in the Abstract and in all text structure. 

2.        I recommend authors to consider the classification of their variant regarding pathogenicity using the ACMG criteria (2015). The current analysis suggests the use of the following criteria: PM2 due to the low frequency of the variant in gnomAD population databases; BS2 due to the observation of the variant in a homozygous state in population databases; PM5 due to the occurrence of different aminoacid change as pathogenic variant; PP2 due to the occurrence of missense variant in this gene with low rate of benign mutations and the common association of missense variants as with deleterious effect; BP6 due to reputable sources recently reporting the variant as benign. The overall evaluation would be to attribute a “likely benign” (class 2) classification of the variant. Furthermore, the variant classification by ClinVar is given as “conflicting”.  

3.        I recommend authors to use the description hATTR to describe hereditary ATTR instead of “ATTRv-PN”. 

4.        I think one interesting point for authors would be to include the discussion about the possibility of wild-type TTR amyloidosis in the presented case. It is of note that the occurrence of marked clinical and neurophysiological improvement after gene silencing therapy does not rule out the possibility of wild-type TTR amyloidosis. I think this point is important to be considered in the discussion even for the context of the final title to be adopted by the authors of the manuscript. 

Author Response

Dear Editors and Reviewers,

Thanks for your comments. We would like to submit our revised version of the manuscript (Manuscript ID: brainsci-2969239 - Encourage Resubmission after Revisions) for possible publication in your journal. Here point-to point answers to the reviewer questions:

#rev 2

  1. I recommend authors to change the use of italics after the TTRgene symbol. The use of italics should be limited to the symbol (i.e., TTR, GSN, SPG11PMP22). This suggestion should be reviewed both in the Abstract and in all text structure. 

A: We thank the reviewer for this suggestion. The manuscript was doublchecked for the correct use of italics for TTR.

  1. I recommend authors to consider the classification of their variant regarding pathogenicity using the ACMG criteria (2015). The current analysis suggests the use of the following criteria: PM2 due to the low frequency of the variant in gnomAD population databases; BS2 due to the observation of the variant in a homozygous state in population databases; PM5 due to the occurrence of different aminoacid change as pathogenic variant; PP2 due to the occurrence of missense variant in this gene with low rate of benign mutations and the common association of missense variants as with deleterious effect; BP6 due to reputable sources recently reporting the variant as benign. The overall evaluation would be to attribute a “likely benign” (class 2) classification of the variant. Furthermore, the variant classification by ClinVar is given as “conflicting”.  

A: We agree with the reviewer. We added these considerations in the discussion.

  1. I recommend authors to use the description hATTR to describe hereditary ATTR instead of “ATTRv-PN”. 

A: We agree with the reviewer. We modified as suggested.

  1. I think one interesting point for authors would be to include the discussion about the possibility of wild-type TTR amyloidosis in the presented case. It is of note that the occurrence of marked clinical and neurophysiological improvement after gene silencing therapy does not rule out the possibility of wild-type TTR amyloidosis. I think this point is important to be considered in the discussion even for the context of the final title to be adopted by the authors of the manuscript. 

A: We agree with the reviewer. We modified as suggested.

“Nonetheless, even if very unlikely, the possibility of wild-type TTR amyloidosis cannot be ruled out in the presented cases as, in very rare cases, the deposition of both wild-type and variant TTR misfolded proteins had been reported”.

Reviewer 3 Report

Comments and Suggestions for Authors

My suggestions:

1. I would add a family history figure for each case, even if the family history was negative. 

2. It is unclear, which patient's data is included in Figure 1. Patient 1 or 2? In the case of both patients, I would separate it, and compare the differences of biopsy between patient 1 and patient 2. 

3. A simple structure prediction for TTN His110Asn may be included in the manuscript. 

4. There is no Methods section. Was there whole genome or whole exome sequencing performed for the patient?

5. Was there whole exome or whole genome sequencing performed on the patient? Were there any other variants related to ataxia observed in the patients?

Author Response

Dear Editors and Reviewers,

Thanks for your comments. We would like to submit our revised version of the manuscript (Manuscript ID: brainsci-2969239 - Encourage Resubmission after Revisions) for possible publication in your journal. Here point-to point answers to the reviewer questions:

#rev 3

My suggestions:

  1. I would add a family history figure for each case, even if the family history was negative. 

A: We thank the reviewer for this suggestion, but we do not agree. In the absence of a family history for ATTRv a family tree will be not contributory.

  1. It is unclear, which patient's data is included in Figure 1. Patient 1 or 2? In the case of both patients, I would separate it, and compare the differences of biopsy between patient 1 and patient 2. 

A: We thank the reviewer for this suggestion. In the revised version, it was specified that the biopsy and all pictures are from patient 1.

  1. A simple structure prediction for TTN His110Asn may be included in the manuscript. 

A: We thank the reviewer for this suggestion, but we are convinced that a structure prediction figure will be not contributory. Moreover, our aim is to describe clinical picture connected to His90Asn, and not data from molecular biology, indeed we chose a clinical journal.

  1. There is no Methods section. Was there whole genome or whole exome sequencing performed for the patient?

A: We thank the reviewer for this suggestion; whole exome sequencing was performed.

  1. Was there whole exome or whole genome sequencing performed on the patient? Were there any other variants related to ataxia observed in the patients?

A: We thank the reviewer for this suggestion. No variants were observed in these two patients.
